# Reproducibility of Training Deep Learning Models for Medical Image Analysis

**Joeran Sander Bosma**[*1]                              Joeran.Bosma@radboudumc.nl
**Dré Peeters**[1]                                       Dre.Peeters@radboudumc.nl
**Natália Alves**[1]                                     Natalia.Alves@radboudumc.nl
**Anindo Saha**[1]                                       Anindya.Shaha@radboudumc.nl
**Zaigham Saghir**[2]                                    zaigham.saghir@gmail.com
**Colin Jacobs**[1]                                      Colin.Jacobs@radboudumc.nl
**Henkjan Huisman**[1]                                   Henkjan.Huisman@radboudumc.nl

[1] *Diagnostic Image Analysis Group, Department of Medical Imaging, Radboud University Medical Center, Nijmegen 6525 GA, The Netherlands*

[2] *Department of Medicine, Section of Pulmonary Medicine, Herlev-Gentofte Hospital, Hellerup, Denmark*

**Editors:** Accepted for publication at MIDL 2023

## Abstract

Performance of deep learning algorithms varies due to their development data and training method, but also due to several stochastic processes during training. Due to these random factors, a single training run may not accurately reflect the performance of a given training method. Statistical comparisons in literature between different deep learning training methods typically ignore this performance variation between training runs and incorrectly claim significance of changes in training method. We hypothesize that the impact of such performance variation is substantial, such that it may invalidate biomedical competition leaderboards and some scientific papers. To test this, we investigate the reproducibility of training deep learning algorithms for medical image analysis. We repeated training runs from prior scientific studies: three diagnostic tasks (pancreatic cancer detection in CT, clinically significant prostate cancer detection in MRI, and lung nodule malignancy risk estimation in low-dose CT) and two organ segmentation tasks (pancreas segmentation in CT and prostate segmentation in MRI). A previously published top-performing algorithm for each task was trained multiple times to determine the variance in model performance. For all three diagnostic algorithms, performance variation from retraining was significant compared to data variance. Statistically comparing independently trained algorithms from the same training method using the same dataset should follow the null hypothesis, but we observed claimed significance with a p-value below 0.05 in 15% of comparisons with conventional testing (paired bootstrapping). We conclude that variance in model performance due to retraining is substantial and should be accounted for.

**Keywords:** Deep learning, reproducibility, medical image analysis, performance variance.

---

[*] Corresponding Author

## 1. Introduction

As the population ages and demographic shifts occur, the healthcare industry is facing an increasing workload (Christensen et al., 2009). To meet this demand, the use of Artificial Intelligence (AI) algorithms, specifically medical image analysis algorithms, is becoming increasingly important. Implementation of such algorithms requires a good understanding of performance on unseen cases. International competitions, so-called (grand) challenges, provide a way to assess and compare the performance of multiple algorithms on identical data sets, providing bias-free evaluation. In challenges, the goal is to identify the best methodology to address a given task, but the difference in performance between top submissions is typically small.

Related research has focused on the repeatability of the AI training method, which requires code sharing, public data, and correct training environment specification (Simkó et al., 2022). We focus on training AI models when these criteria are met, investigating the performance reproducibility of training deep learning-based medical image analysis algorithms. Performance of deep learning models can vary significantly due to the stochastic processes during training, as demonstrated in previous studies for natural image classification and segmentation, natural language modeling, and regression with a multi-layer perceptron (Bouthillier et al., 2021; Picard, 2021; Summers and Dinneen, 2021; Frankle and Carbin, 2018; Reimers and Gurevych, 2017). This variability makes it challenging to accurately assess the performance of a deep learning training method, because retraining the same algorithm on the same dataset results in a different performance each time. In particular, for deep learning in medical imaging, comparisons with radiologists or alternative training methods can vary widely depending on the specific training run. If this variance is large, the conclusions of non-inferiority compared to radiologists or superiority of methodological changes in a training pipeline can depend on the specific training run.

By comparing model performance between different training runs, we aim to investigate the reproducibility of training deep-learning models for medical image analysis. We consider three diagnostic tasks: pancreatic cancer (PaC) detection in CT (Alves et al., 2021), clinically significant prostate cancer (csPCa) detection in biparametric MRI (Bosma et al., 2021), and lung nodule malignancy (LNm) risk estimation in low-dose CT (Venkadesh et al., 2021). These tasks are clinically relevant, span both MR and CT, and include 3 out of the 8 most deadly cancers (Sung et al., 2021). We included two organ segmentation tasks: pancreas segmentation in CT (Alves et al., 2021) and whole-gland prostate segmentation in biparametric MRI (Saha et al., 2022). Previously published top-performing algorithms are trained multiple times on the same data, to determine the variation in performance due to random weight initialization, random data sampling, and random data augmentation.

Understanding performance variation is crucial to identify technological improvements in training pipelines, as it allows us to differentiate the added benefit of proposed methodologies from stochastic factors (e.g. "lucky" weight initialization). This is particularly relevant in the context of biomedical competitions, so-called (grand) challenges. In challenges the difference in performance between top submissions is typically small, and the ultimate goal is to identify the best methodology to address a given task. By understanding the variance in performance of deep learning-based medical image analysis algorithms, we can make more informed conclusions about effectiveness, and guide future research in this area.

## 2. Materials and Methods

### 2.1. Datasets and Algorithms

For the task of csPCa detection, the datasets consisted of biparametric MRI (axial T2-weighted, high b-value diffusion-weighted imaging, and apparent diffusion coefficient maps). The training dataset consisted of 3050 cases (1080 malignant) with 1,315 PI-RADS $\geq 4$ lesions. The test dataset contained 300 cases (88 malignant) from an external institution, with 97 histopathology-confirmed malignant (ISUP $\geq 2$) lesions. The algorithm used was a previously established deep learning-based method that takes an MRI scan as input and produces lesion candidates with malignancy likelihood scores as output. This algorithm was trained using the nnU-Net framework (Isensee et al., 2020) and consists of an ensemble of 5 models, obtained by training 3D U-Net models with 5-fold cross-validation and cross-entropy loss. The algorithm extracted lesion candidates with malignancy scores from the softmax predictions. See (Bosma et al., 2021) for additional details on the dataset and algorithm.

For the task of PaC detection, the datasets consisted of contrast-enhanced CT scans in the portal-venous phase. The training dataset consisted of 242 cases (119 malignant) with 119 histopathology-confirmed malignant lesions. The test dataset contained 361 cases (281 malignant) from two external institutions, with 281 histopathology-confirmed malignant lesions (Simpson et al., 2019; Roth et al., 2016). The algorithm used was a previously established deep learning-based method that takes a CT scan as input and produces lesion candidates with malignancy likelihood scores as output. This algorithm was trained using the nnU-Net framework and consists of an ensemble of 5 models, obtained by training 3D U-Net models with 5-fold cross-validation and cross-entropy loss. The algorithm extracted lesion candidates with malignancy scores from the softmax predictions. See (Alves et al., 2021) for additional details on the dataset and algorithm.

For the task of LNm risk estimation, the datasets consisted of low-dose screening CT scans. The training dataset consisted of 5282 cases (686 malignant) with 16077 lesions (1249 malignant). The test set contained 602 cases (62 malignant) from an external institution, with 883 lesions (65 malignant). For each dataset, the reference standard was set by histopathological confirmation for malignant lesions, and at least 2-year stability for benign lesions. The algorithm used was an ensemble of 20 models, obtained by training one 2D and one 3D convolutional neural network, each with 10-fold cross-validation. This algorithm is a deep learning-based method that takes a CT scan and the nodule coordinates as input and produces a malignancy likelihood score as output and has been previously reported in (Venkadesh et al., 2021). See (Venkadesh et al., 2021) for additional details on the dataset and algorithm.

For prostate segmentation, the datasets consisted of biparametric MRI scans. The training dataset consisted of 198 cases from the ProstateX dataset (Armato III et al., 2018) with prostate segmentations from experienced radiologists (Cuocolo et al., 2021), as well as 101 institutional cases. Testing was performed with 139 cases with prostate segmentations by experienced radiologists from the Prostate158 training dataset (Adams et al., 2022). The algorithm used was a vanilla nnU-Net algorithm that takes an MRI scan as input and produces a prostate segmentation as output. This algorithm was trained using the nnU-Net

framework and consists of an ensemble of 5 models, obtained by training 3D U-Net models with 5-fold cross-validation. The preprocessing pipeline was taken from (Saha et al., 2022).

For pancreas segmentation, the same dataset and training method were used as for PaC detection. Unlike the algorithm for PaC detection, the model checkpoints in the ensemble were the final checkpoints, as per the default for segmentation in the nnU-Net framework.

## 2.2. Evaluation

Malignancy-detection algorithms were evaluated at both case-level diagnosis and lesion-level detection. The case-level diagnosis performance was evaluated using the Area Under the Receiver Operating Characteristic (AUROC) metric. To evaluate the lesion-level detection performance of csPCa or PaC, the 3D detection maps of non-overlapping, non-connected lesions with an associated malignancy-risk score were evaluated using the Average Precision (AP) metric. True positives required a minimum overlap of 0.10 Intersection over Union (IoU) in 3D with the ground-truth annotation. False positives were defined as predictions with no or insufficient overlap, regardless of size or location. When multiple predictions had sufficient overlap, only the prediction with the largest overlap was counted as a true positive, while all other overlapping predictions were discarded. An overview of this evaluation pipeline is shown in Figure 1. To evaluate the lesion-level risk estimation performance of LNm, the AP and lesion-level AUROC were calculated for the predefined set of lesions. Segmentation performance was evaluated using the Dice similarity coefficient (DSC).

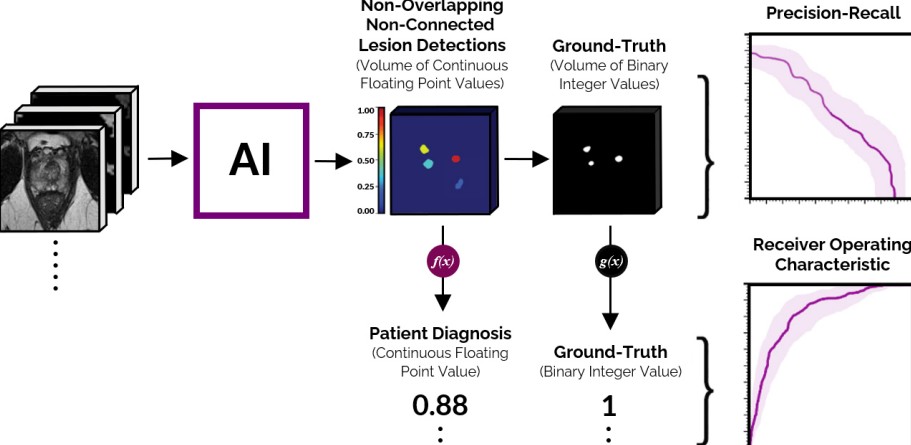

Figure 1: Evaluation strategy for detection and diagnosis of csPCa or PaC. (Top) Lesion-level detection involves predicting a 3D map of non-overlapping, non-connected lesions with a likelihood score for each lesion. (Bottom) Case-level risk score is computed using the predicted lesion detection map. Lesion-level detection maps and case-level risk scores are compared with the ground truth to evaluate the algorithm.

### 2.3. Statistical Analysis

Many studies employ paired bootstrapping to estimate the probability that a given training method outperforms an alternative training method. When we perform this statistical test between different training runs of a given training method, the null hypothesis holds by definition (the performance is not different, since the models are trained with the same training pipeline). As such, we can investigate the calibration of paired bootstrapping by looking at the p-values from paired bootstrapping between training runs in our experiments. If a statistical test is well-calibrated, the p-values obtained from comparisons under the null hypothesis should be uniformly distributed (i.e, 5% of the comparisons should have a p-value below 0.05, 1% of the comparisons should have a p-value below 0.01, etc.). We check whether paired bootstrapping is well-calibrated by comparing different training runs from the same training pipeline. See Appendix B for details.

To compare training methods with empirical rigor, the performance variance due to the stochasticity in the training pipeline should be accounted for. To achieve this, multiple instances must be trained to construct the performance distribution of models trained with a given training pipeline. The probability of one training method outperforming another training method can then be computed using a permutation test with the performance metrics, as done in (Bosma et al., 2021) and described in Appendix C. This accounts for the variance in model performance across training runs, and the number of training runs (sample size) supporting the comparison. We check whether the permutation test with performance metrics is well-calibrated by performing simulations, as detailed in Appendix C.

### 2.4. Computational Resources

Training deep learning-based algorithms for medical image analysis is typically expensive. We allowed a computational budget of approx. 4000 GPU hours on an RTX 2080 Ti to train all methods. Training a nnU-Net ensemble with 5-fold cross-validation took approx. 180 GPU hours. Training the ensemble for LNm risk estimation took approx. 20 GPU hours. We decided to train the nnU-Net ensembles 7 times, and the LNm ensemble 10 times, to fit in the allotted computational budget. For computational feasibility we implemented the following changes with respect to the original publication:

- Ensemble 5 models (single 5-fold cross-validation iteration) for csPCa detection, instead of 15 models (three 5-fold cross-validation iterations) as in (Bosma et al., 2021),

- Ensemble 5 models (single 5-fold cross-validation iteration) for PaC detection, instead of 10 models (two 5-fold cross-validation iterations) as in (Alves et al., 2021).

### 3. Results

Figure 2 shows the performance distribution of the five deep learning-based image analysis algorithms investigated in this study. Each algorithm is an ensemble of 5 or 20 models, and performance differences are a result of retraining the full ensemble, due to random weight initialization, random data sampling, and random data augmentation. The results show substantial variance in diagnostic performance: at case-level diagnostic performance

(as measured by AUROC), csPCa diagnosis differed 2.3% between best and worst run, PaC diagnosis differed 3.8% between best and worst run, and LNm diagnosis differed 1.7% between best and worst run. Variance in segmentation performance was comparable for prostate segmentation: DSC differed 1.9% between best and worst run. For pancreas segmentation, the DSC differed only 0.2% between the best and worst run. In Appendix A an overview of model performance variation is shown.

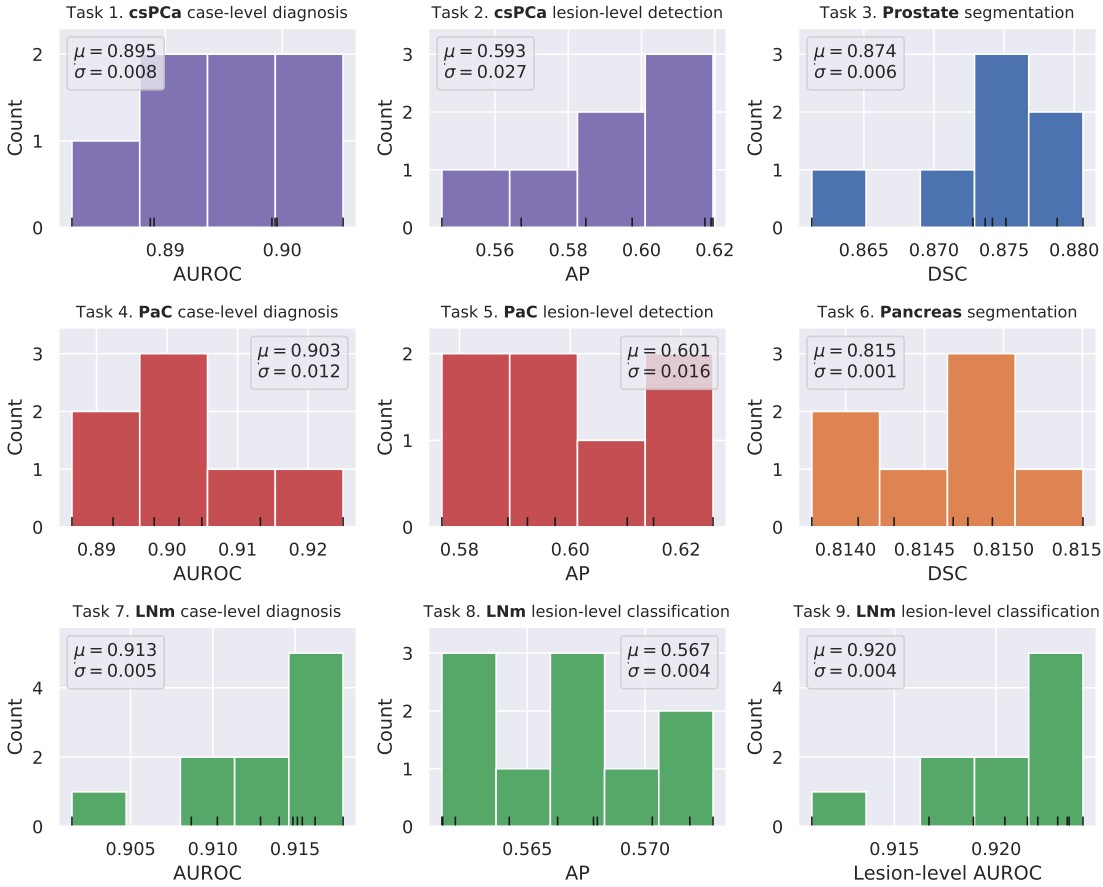

Figure 2: Distribution of performances across training runs for clinically significant prostate cancer (csPCa) detection, pancreas cancer (PaC) detection, lung nodule malignancy (LNm) risk estimation, prostate segmentation and pancreas segmentation. Malignancy-detection algorithms are evaluated at both case-level diagnosis (AUROC) and lesion-level detection (AP). The x-axis represents the performance metric, while the y-axis shows the frequency of performance within a particular range. For lesion-level LNm classification, we included the lesion-level AUROC, following (Venkadesh et al., 2021). For segmentation, we show the distribution of Dice similarity coefficient (DSC) scores.

Performing image analysis with ensembles of 5-20 models is not feasible in certain clinical settings. For example, inference speed is essential for radiotherapy planning, real-time tracking, or when compute is provided by a CPU. To aid radiologists, predictions need to be presented quickly to achieve a good workflow, but cannot always be pre-computed. Therefore, we show the performance distribution for individual models in Appendix A.

Paired bootstrapping claimed a p-value below 0.05 in 26/174 (15%) comparisons between different trained ensembles. This means that paired bootstrapping claimed statistical significance of a difference in model performance $3\times$ more often than it should. For lower p-values this mismatch was even worse: 3/174 (1.7%) comparisons had a p-value below 0.001, a miscalibration of $17\times$. See Appendix B for details.

In our simulations with the permutation test, we observed a p-value below 0.05 in $1,418/30,000$ (4.7%) comparisons. This means that the permutation test claimed statistical significance of a difference in model performance almost perfectly as often as it should. At lower p-values the calibration remained very good: $292/30,000$ (1.0%) of the comparisons had a p-value below 0.01, a near-perfect calibration. We observed a p-value below 0.001 for $20/30,000$ (0.07%) comparisons, claiming statistical significance $1.5\times$ less often than it should (i.e., a bit conservative). See Appendix C for details.

## 4. Discussion

Diagnostic performance of the algorithms varied significantly between training runs, even though the training and evaluation datasets were fixed and each algorithm was an ensemble of at least 5 models. Performance variation due to random weight initialization, random data augmentation, and random data order obscures the true performance of the training method when training only a single instance. As a result, researchers could perceive random fluctuations in performance as evidence for methodological changes in the training pipeline, potentially moving AI researchers in sub-optimal directions.

Prostate segmentation performance varied considerably between training runs, while pancreas segmentation performance was very consistent. This may be a result of the underlying modality: the MRI scans for prostate segmentation have lower resolution than the CT scans for pancreas segmentation, resulting in a larger segmentation difference when in/excluding a single slice in the segmentation. However, more segmentation tasks need to be investigated to obtain a better understanding of performance variance between training runs for segmentation algorithms.

Our experiments provide evidence that paired bootstrapping is not suitable for estimating the probability that a given training method outperforms an alternative training method. On the other hand, the permutation test with performance metrics accounts for performance variation in the training method. We showed that the permutation test is well-calibrated, and therefore valid to statistically compare training methods with empirical rigor. However, comparing training pipelines using the permutation test can be computationally expensive. Medical image analysis algorithms are often ensembles, which must be trained $5 - 10\times$ to obtain sufficient statistical power. Even if this computational power is available, the question remains whether the environmental impact of the additional computational resources is justified by the additional empirical rigor.

Multiple training runs allow generalizing experimental results to the training method, rather than the trained algorithm. However, the experimental outcomes are still specific to the task, development dataset, test dataset, and evaluation metric. Changes in these factors can change the outcome of experiments (Maier-Hein et al., 2018), so claims cannot be generalized without accounting for these factors. We recommend authors to carefully think about which claims can be made based on the performed experiments, as these claims are likely limited to the specific trained algorithm, task, datasets, and evaluation method.

Within challenges, the top submissions often achieve performances in a narrow range. Within one of our experiments, we performed the same task as done in the PI-CAI Challenge (csPCa detection), using similar data and with a comparable test set and evaluation strategy. Between training runs of our algorithm at this task, we observed a standard deviation for the PI-CAI Ranking Score[1] of $\sigma = 0.013$ between training runs. If we assume this standard deviation for each submission of the PI-CAI Challenge, the probability that the current #1 submission would place #1 again when each method is retrained, is only 38%. See Appendix D for details. Challenges with multiple datasets (e.g., the Medical Segmentation Decathlon with 3 test datasets (Antonelli et al., 2021)) evaluate multiple runs, implicitly accounting for training variance to some degree. However, our results suggest that leaderboard rankings in typical medical image analysis challenges are not suitable to find the best training method, and challenge leaderboards should be interpreted with this nuance.

It is important to note that depending on the goal of a research project or (grand) challenge, the performance variance due to stochasticity in the training pipeline may or may not be important. When developing an algorithm for use in clinical routine, the performance of the trained algorithm matters, not the stability of the training method. Additionally, for clinical deployment, factors such as robustness, interpretability, inference speed, uncertainty quantification, and out-of-distribution detection are key. On the other hand, when benchmarking training methods, introducing a new training pipeline, investigating preprocessing or data augmentation techniques, or changing the dataset (e.g., comparing supervised with semi-supervised learning, adding a new model input, or validating a reconstruction/super-resolution model), accounting for the performance variation during training is crucial. This variance must be accounted for because the downstream performance of the trained algorithms is used to infer the effect of changes made prior to or within the training pipeline.

Despite the insightful findings, there are some limitations to this study. First, it was unfortunately not possible to exactly reproduce the algorithms of the original publications. The main reason was computational resource availability, as the original publications applied ensembles with $10 - 20$ independent models. Although this is not ideal, the introduced changes were minor, and the algorithms remain top-performing and clinically relevant. Secondly, more tasks could have been investigated. The set of tasks we chose is clinically relevant and covers a wide variability of medical image analysis tasks, but could be improved by including more types of tasks. Finally, we note that the manual pancreas segmentations of the external test set are poor, which limits the ability to evaluate segmentation performance accurately. This is important to consider when interpreting the pancreas segmentation results, as it suggests that this experiment may not be as reliable as the other tasks.

---

1. The PI-CAI Ranking Score is the average of the case-level diagnostic AUROC and lesion-level AP.

## 5. Conclusion

Diagnostic performance of the deep learning-based medical image analysis algorithms varied significantly between training runs, while deep learning-based segmentation performance was stable. The diagnostic performance varied due to random initialization, random data augmentation, and random data order, obscuring the true performance of training pipelines when only a single instance is trained. Comparing training pipelines with paired bootstrapping leads to over-claiming statistical significance of performance differences, mistaking random fluctuations in performance for technological improvements. The findings of this study demonstrate the importance of understanding the performance variance when training deep learning models and show that diagnostic algorithms need to be evaluated with empirical rigor.

## Acknowledgments

We thank Ivan Slootweg for his efforts in annotating the dataset for csPCa detection. We thank Kiran Vaidhya Venkadesh for providing the training pipeline for LNm risk estimation.

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

## Appendix A. Performance Distribution Individual Models

When evaluating the models from the ensembles individually, we observe the distributions shown in Figure 3. For each task, the variation in performance is larger compared to the variation in performance when evaluating ensembles. In Figure 4 an overview of model performance variation is shown.

## Appendix B. Calibration of Paired Bootstrapping

Paired bootstrapping is a statistical method that is often used to estimate the probability that a given training method outperforms an alternative training method. It involves re-sampling pairs of observations from a dataset and using the resampled data to estimate the difference between the performance of the two methods being compared.

You would use the paired bootstrapping procedure to generate a large number of re-sampled datasets by sampling pairs of observations from the test dataset and calculating the difference in performance between the two algorithms on each resampled dataset. This process is repeated many times to generate a distribution of performance differences.

Then, you can use this distribution to calculate the probability that the first method outperforms the second. This can be done by counting the number of times that the performance difference in the resampled datasets is greater than zero (indicating that the first method outperformed the second) and dividing by the total number of resampled datasets. The resulting probability can be interpreted as an estimate of the likelihood that the first method is truly superior to the second.

We can investigate whether paired bootstrapping is suitable to estimate the probability that the first method is truly superior to the second. When we perform this statistical test between different training runs of an algorithm, the null hypothesis holds by definition (the performance is not different, since the models are trained with the same training pipeline). As such, the p-values obtained from these comparisons should be uniformly distributed (i.e, 5% of the comparisons should have a p-value below 0.05, 1% of the comparisons should have a p-value below 0.01, etc.) (Dawid, 1982). We calculated the p-value from paired bootstrapping for each combination of training runs for case-level csPCa diagnosis, PaC diagnosis, or LNm diagnosis, giving $n^2 - n$ comparisons per use case.

Comparing ensembles from independent training runs, yielded 42 comparisons for csPCa diagnosis, 42 for PaC diagnosis, and 90 for LNm diagnosis, so 174 comparisons in total. We did not include pancreas segmentation, because those training runs were shared with PaC diagnosis. The p-values from paired bootstrapping for each comparison are given in Figure 5. We observed a p-value below 0.05 in 26/174 (15%) comparisons. This means that paired bootstrapping claims statistical significance of a difference in model performance 3× more often than it should. At lower p-values this mismatch was even worse: 12/174 (6.9%) of the comparisons had a p-value below 0.01, a miscalibration of 6.9×. We observed a p-value below 0.001 for 3/174 (1.7%) comparisons, claiming statistical significance at that threshold 17× more often than it should.

Comparing individual models from independent training runs, gives 1,190 comparisons for csPCa detection, 1,190 for PaC detection, 9,900 for LNm classification with a 2D model and 9,900 for LNm classification with a 3D model, so 22,180 comparisons in total. The observed p-values are shown in Figure 6. We observed a p-value below 0.05 for $3,534/22,180$

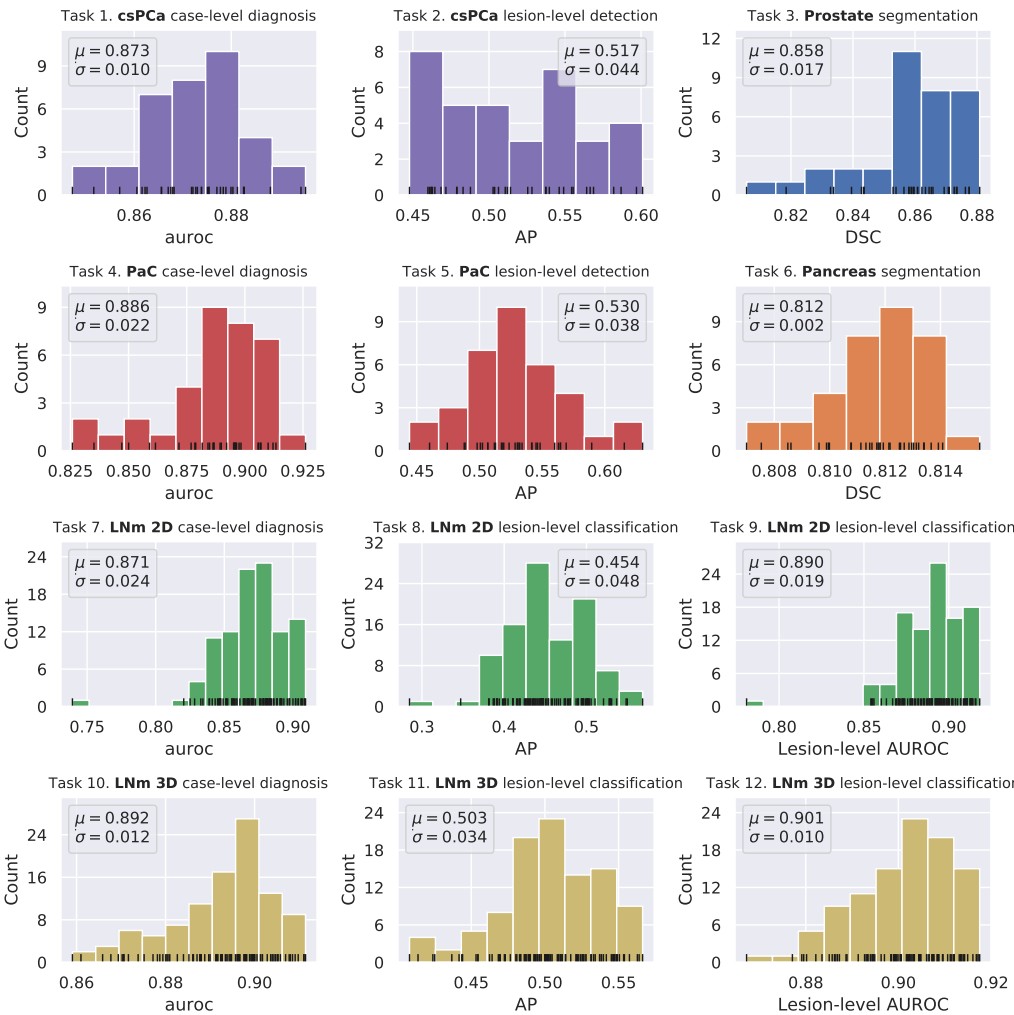

Figure 3: Distribution of performances across training runs for clinically significant prostate cancer (csPCa) detection, pancreas cancer (PaC) detection, lung nodule malignancy (LNm) risk estimation, prostate segmentation and pancreas segmentation. Individual models are evaluated, rather than ensembles. Malignancy-detection algorithms are evaluated at both case-level diagnosis (AUROC) and lesion-level detection (AP). The x-axis represents the performance metric, while the y-axis shows the frequency of performance within a particular range. For lesion-level LNm classification, we included the lesion-level AUROC, following (Venkadesh et al., 2021). For segmentation, we show the distribution of Dice similarity coefficient (DSC) scores.

(16%) comparisons, claiming statistical significance $3.2\times$ too often. At lower p-values, this became worse, with $1,765/22,180$ (8.0%) comparisons below 0.01, a miscalibration of $8.0\times$. Finally, we observed $665/22,180$ (3.0%) comparisons with a p-value below 0.001,

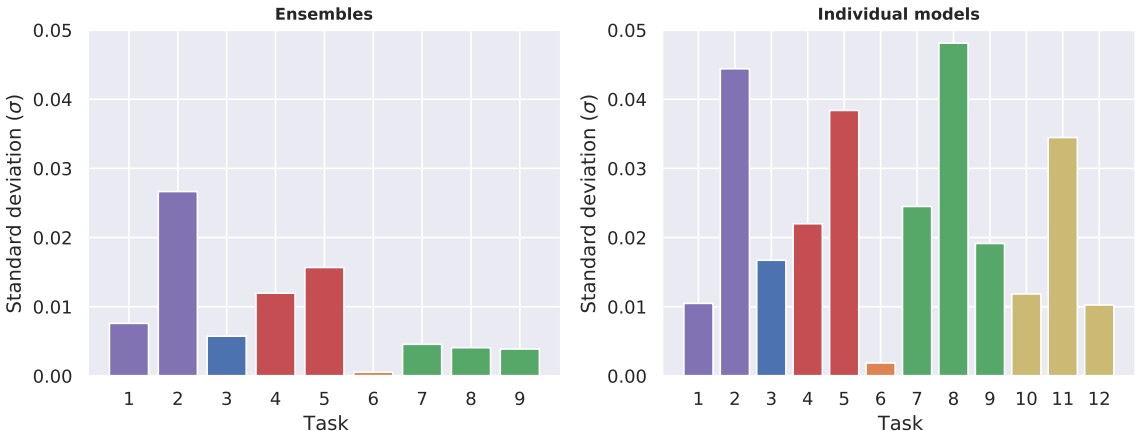

Figure 4: Performance variability between training runs. The left panel shows the ensembles (containing 5-20 models) presented in Figure 2 and the right panel shows the individual models presented in Figure 3.

claiming statistical significance at that threshold $30\times$ too often. The calibration of paired bootstrapping is similar when comparing individual models as when comparing ensembles.

The Wald interval for binomial distributions allows to calculate the confidence interval for the observed outcomes of the paired bootstrapping test (Wallis, 2013), as follows:

$$\hat{p} \pm z\sqrt{\frac{\hat{p}\left(1-\hat{p}\right)}{n}} \tag{1}$$

Where $\hat{p} = x/n$, $n$ the total number of trails, $x$ the number of times the event occurred, and $z$ the quantile of a standard normal distribution corresponding to the target confidence interval. For the 95% confidence interval, $z$ is 1.96. Given 26 observations of the event "p-value claimed to be below 0.05" in 174 trails, the 95% Wald confidence interval is $(0.0964 - 0.2024)$. Since we know that p-values below 0.05 should only occur 5% of the times, which does not fall within the 95% confidence interval, we can reject that paired bootstrapping is valid for comparing training pipelines.

Our experiments provide evidence that paired bootstrapping is not suitable for estimating the probability that a given training method outperforms an alternative training method. This makes intuitive sense because the variation in performance due to the training pipeline is fundamentally different from the variability estimated from bootstrapping the likelihood table (i.e., variance stemming from the evaluation dataset). There is no statistical reason why those two sources of variance should be comparable.

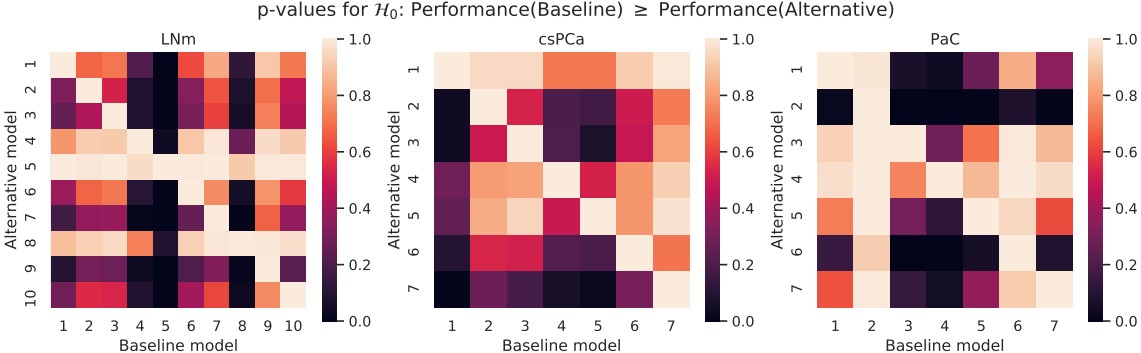

Figure 5: P-values from paired bootstrapping for each comparison of training runs for case-level csPCa diagnosis, PaC diagnosis, and LNm diagnosis. Each algorithm is an ensemble of $5 - 20$ models.

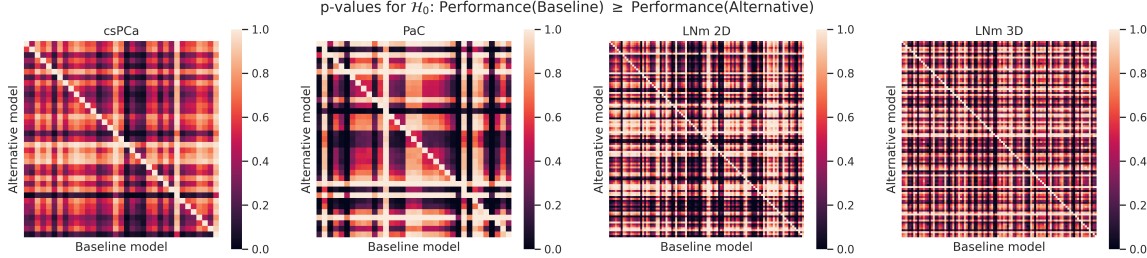

Figure 6: P-values from paired bootstrapping for each comparison of training runs for case-level csPCa diagnosis, PaC diagnosis, and LNm diagnosis using the individual models (i.e., the models that are part of the ensembles).

## Appendix C.  Calibration of Permutation Test

A permutation test is a nonparametric statistical test that can be used to determine whether the difference between two groups is statistically significant. It is often used when the assumptions of parametric tests, such as the t-test or ANOVA, are not met. To conduct a permutation test, the following steps are typically followed:

1. Calculate the test statistic for the observed data.

2. Randomly permute the data and recalculate the test statistic.

3. Repeat step 2 a large number of times (e.g., 1000) to generate a distribution of the test statistic under the null hypothesis.

4. Calculate the p-value as the proportion of permuted test statistics that are equal to or more extreme than the observed test statistic.

The permutation test is useful because it does not rely on assumptions about the underlying distribution of the data. It is also relatively simple to implement, as it only requires the calculation of the test statistic and the ability to randomly permute the data. To account for the variability of model performance between training runs, the observed performance metric from each independent training run should be permuted. This type of permutation test is implemented in picai_eval. We use this implementation in our simulations.

The permutation test requires an order of magnitude more compute resources, so empirically checking its calibration is infeasible. However, because the permutation test is not an approximation (like paired bootstrapping), it is inherently valid, and simulations can easily show that it is well-calibrated. To check whether the permutation test when permuting performance metrics is valid to compare training pipelines, we simulate a large number of training runs. In our simulation, we generate a random set of predictions as follows:

$$y_{pos} \sim \text{Beta}(\alpha = 3, \beta = 2) \text{ and } y_{neg} \sim \text{Beta}(\alpha = 2, \beta = 6) \tag{2}$$

This gives a distribution of predictions for positive and negative cases as shown in Figure 7. When sampling 300 positive cases and 700 negative cases, these predictions have a case-level classification AUROC of $0.91 \pm 0.01$. Sampling these sets of predictions is quick, so we can generate sets of predictions with the same parameters to perform many permutation tests under the null hypothesis. We sample 10 sets of predictions for the baseline training method (simulating retraining the algorithm 10 times) and 10 sets of predictions for the alternative training method, to perform the permutation test with a lot of power. We performed the permutation test with 3,000 random permutations and simulated $30,000$ comparisons.

The calibration of p-values from the permutation test and paired bootstrapping is shown in Figure 8. In our simulations, we observed a p-value below 0.05 in $1,418/30,000$ ($4.7\%$) comparisons. This means that the permutation test claimed statistical significance of a difference in model performance almost perfectly as often as it should. At lower p-values the calibration remained very good: $292/30,000$ ($1.0\%$) of the comparisons had a p-value below 0.01, a near-perfect calibration. We observed a p-value below 0.001 for $20/30,000$ ($0.07\%$) comparisons, claiming statistical significance $1.5\times$ less often than it should (i.e., a bit conservative).

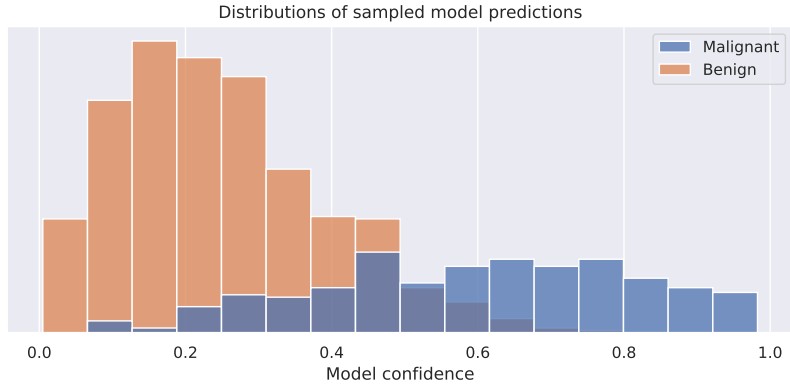

Figure 7: Distributions of model predictions, with predictions for positive (malignant) and negative (benign) cases sampled according to Equation (2).

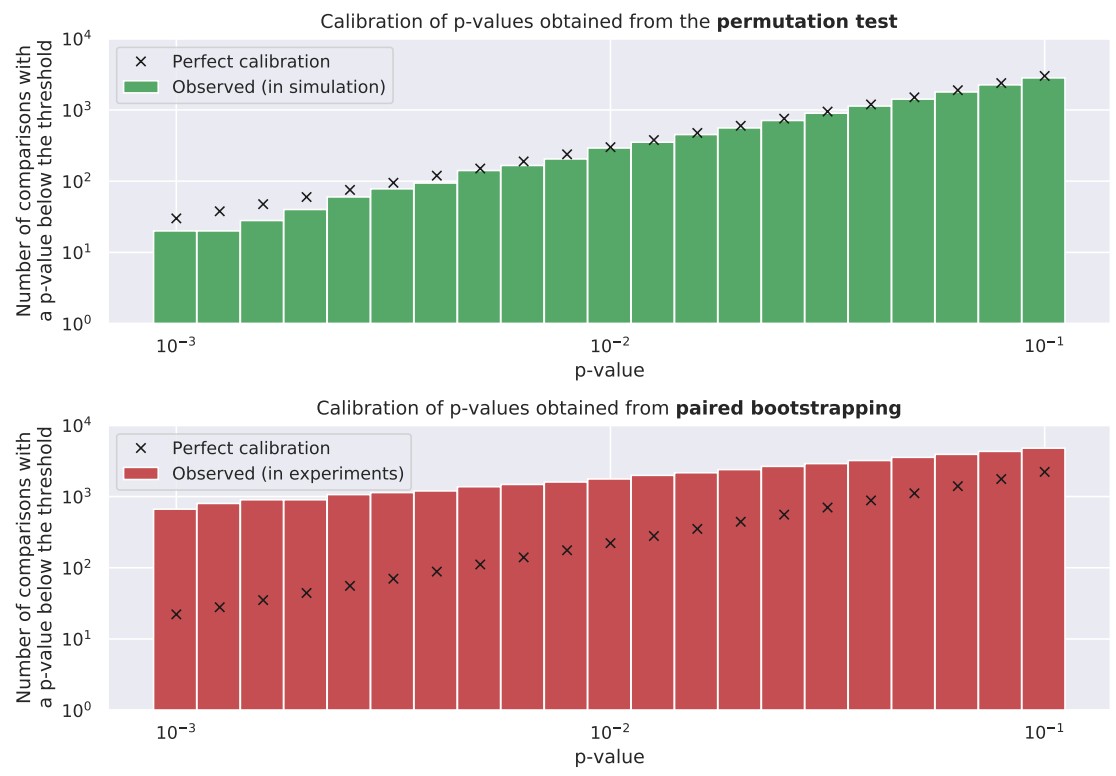

Figure 8: Calibration of p-values obtained from (top) the permutation test with simulated model performances, and (bottom) paired bootstrapping with individual models. $30,000$ comparisons were simulated for the permutation test, and comparing individual models provided $22,180$ comparisons for paired bootstrapping. The crosses indicate the number of comparisons with a p-value below a given threshold a perfectly calibrated test has when the null hypothesis holds.

## Appendix D. Challenge Winner Simulation

Challenges often attract many teams, and the performances obtained by top-performing teams are often close. See Table 1 for an overview of representative challenges in the medical image analysis field. If the variation in model performance is large, the order of the submissions could change if each method was re-trained.

We can estimate the probability that any of the submissions would have won the challenge, by simulating re-training each method many times. For this, we assume that the observed performance for each submission is drawn from a Gaussian distribution, where we set the mean of the distributions to the observed performance in the challenge. Furthermore, we assume that each Gaussian distribution shares the same standard deviation. Under these assumptions, the win probability can be estimated for each method using Algorithm 1. We estimated the win probability of the nr. 1 submission for each challenge in Table 1, and show the win probabilities in Figure 9.

Within our experiments, we performed the same task (csPCa detection) as the PI-CAI Challenge, using the same kind of data (biparametric MRI, except for the clinical information) and evaluated performance in the same way (using picai_eval) against a very similar reference standard (histopathology-confirmed malignant lesions). We observed a standard deviation for the PI-CAI ranking score $= (\mathrm{AUROC} + \mathrm{AP})/2$ of $\sigma = 0.013$ between training runs. At this standard deviation, the win probability of the nr. 1 submission is only 38%.

Similarly, for the AIROGS (Referable glaucoma pAUC) challenge, which had the median $\Delta(1-3)$, the win percentage at the same $\sigma = 0.013$ was 45%.

```python
import numpy as np
scores = [0.757, 0.752, 0.752, 0.742, 0.740]
std_options = np.logspace(-3, -1, num=101)
win_rates = {i: {} for i in range(len(scores))}

for std in std_options:
    simulations = np.zeros(100_000)

    for i in range(len(simulations)):
        observed_performance = [
            np.random.normal(loc=score, scale=std)
            for score in scores
        ]
        simulations[i] = np.argmax(observed_performance)

    for i in range(len(scores)):
        win_rates[i][std] = np.sum(simulations==i) / len(simulations)
```

**Algorithm 1:** Estimate win probability of challenge submissions

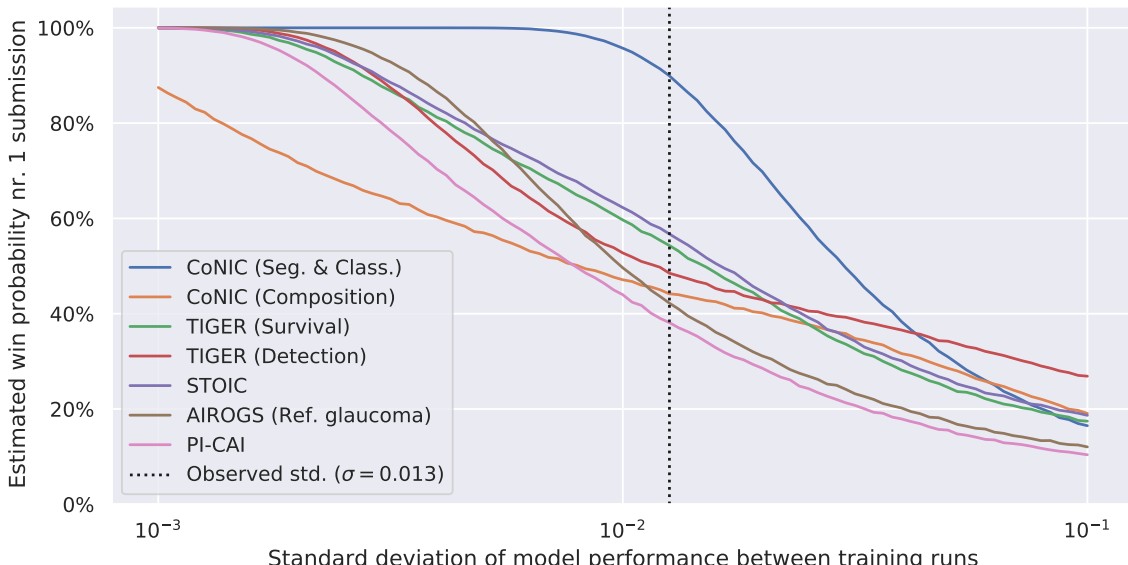

Figure 9: Probability that the nr. 1 submission would win the challenge again when each method is retrained, assuming the same standard deviation in model performance between training runs for each method. Several Grand Challenges are presented: CoNIC, TIGER, STOIC, AIROGS and PI-CAI. At the standard deviation observed in our experiments ($\sigma = 0.013$), the probability of the nr. 1 submission to win again is about 50%.

Table 1: Top 3 performances for challenges highlighted on the Grand Challenge blogs, and the challenge closest to the investigated malignancy detection use cases (PI-CAI). See the respective challenge website for details on the task, metrics, and datasets. $\Delta(1-3)$ is the difference in performance between the nr. 1 and nr. 3 submissions.

| Challenge | Nr. 1 | Nr. 2 | Nr. 3 | $\Delta(1-3)$ |
|---|---|---|---|---|
| CoNIC (Segmentation and Classification) | 0.5013 | 0.4762 | 0.4631 | 0.0382 |
| CoNIC (Composition) | 0.7641 | 0.7625 | 0.7550 | 0.0091 |
| TIGER (Survival) | 0.6388 | 0.6338 | 0.6224 | 0.0164 |
| TIGER (Detection) | 0.5504 | 0.5441 | 0.5437 | 0.0067 |
| STOIC | 0.8154 | 0.8100 | 0.7943 | 0.0211 |
| AIROGS (Referable glaucoma pAUC) | 0.8998 | 0.8932 | 0.8873 | 0.0125 |
| PI-CAI | 0.757 | 0.752 | 0.752 | 0.0050 |

