# OpenReview forum: "Reproducibility of Training Deep Learning Models for Medical Image Analysis"
_MIDL.io/2023/Conference — MIDL 2023 Poster_

### Official Review · Reviewer_iJYy · 2023-02-04

**Confidence:** 2
**Preliminary Rating:** 4
**Recommendation:** Poster

**Summary:**

Using four medical image analysis problems (pancreas cancer detection from CT, prostate cancer detection from MRI, lung nodule malignancy scoring from CT, and pancreas segmentation from CT), the authors train several times the same system and study performance variation across training runs. Results reveal that

**Strengths:**

- I don't think the findings here are very surprising for anyone who has ever trained a deep learning system more than once, but it is nice to see this explicitly exposed with statistical rigour.
- There is a variety of scenarios and problems studied, under different image modalities.
- I find it quite interesting that the winner of the PI-CAI challenge may just simply got lucky in a coin toss with 38% probability of success (!!).

**Weaknesses:**

- Even if the amount of models trained is considerabe, because the authors use five-fold ensembles (à la nnUnet), the actual test set predictions are not so many. I wonder if maybe if would have been more reasonable to train smaller models, or in smaller problems, or not using ensembles, to gain statistical power.
- In the end, the authors point to a (well-known) problem and confirm that it exists, but do not propose any constructive mechanism to handle this problem.

**Deanonymize Review:**

no

**Detailed Comments:**

I must admit I do not have (or have forgotten) the required knowledge in statistics to understand calibration of permutation tests, so I do not feel qualified to judge that part of the paper. The remaining of the work seems to me well-written and rigorous.

There is a last year MIDL paper (https://openreview.net/forum?id=7Cov6FwmOP1) that seems to me is quite related to what this paper deals with (I might be wrong). Maybe the authors could discuss this kind of efforts as potential solutions?


**Paper Type:**

validation/application paper

**Questions To Address In The Rebuttal:**

I understand that the authors are using models they already trained in previous works of them (all the "redacted" references I'm guessing). However, wouldn't it be more logical to use smaller datasets to train a reasonably performing model more times on the same data split? Why was this not the initial strategy?

In addition, ensembling probably stabilizes predictions and reduces variance, and not all of us always uses ensembles because they are quite expensive, so I guess that using single-fold models would have been justificed and woudl have probably exposed the underlying repeatibility problem in a much more impressive manner.

---

### Official Review · Reviewer_EC5r · 2023-02-06

**Confidence:** 4
**Preliminary Rating:** 4
**Recommendation:** Poster

**Summary:**

In this work, the authors investigate reproducibility of deep learning(DL) algorithms for different medical image analysis tasks. Authors state that the DL methods performance can vary due to different stochastic processes involved in training like random weight initialization, data sampling and augmentations. This can be problematic in clinical deployment scenarios when the variations between training runs of DL models is huge and thereby, the conclusions cannot be relied upon. They evaluate this on 3 diagnostic tasks of pancreatic cancer, prostate cancer, lung nodule malignancy risk estimation and 1 segmentation task of pancreas. The evaluation is done by performing multiple runs on the same dataset to capture variation between runs.


**Strengths:**

1. The article is well-written and easy to follow.
2. The experimentation and its details of the datasets and model runs along with the presentation of results is clear.
3. The method paired bootstrapping and relevant methodology used for evaluation is well described in the main manuscript and required additional details are provided in the appendix.


**Weaknesses:**

1. In Fig 2. presenting the results, the standard deviation (wrt 7/10 runs) for most tasks is around 0.01. I believe this is a reasonable reproducibility for a deep learning algorithm to deploy in a clinical use case but I agree that this variation does affect the challenge rankings (as the authors mention this in the discussion section). Do the authors think that this is a major concern in performance that would be detrimental to deploy in clinical scenarios?
2. Can the authors provide the top performing algorithm scores for each challenge dataset from the official challenge website? It would be interesting to compare it against the score obtained with the best performing run of the authors in this work.
3. The authors mention that segmentation score on the test set of pancreas is reproducible compared to detection task. Later, they mention that the algorithm does not perform well on an external dataset. I believe this is an out of distribution problem and does not go well with the current article. This would most likely happen if the authors test on an external dataset for the detection tasks as well. Can the authors edit this part in the discussion section to keep it in flow with the current scope of the article or justify why they believe they need to include this?
4. The authors mention that the submission rank of the top performing algorithm on the challenge would go down for the observed standard deviation. Can the authors elaborate a bit more on this and provide some sample top 3 scores as it will be easy to comprehend this point for the readers.
5. In section 2.3, the authors mention the calibration of paired bootstrapping and come back to this discussion in results section 3 on page 5. Currently, it is not clear how the authors use it for their hypothesis testing and computation of p-value and what does 174 indicate here?
6. In computation resources, the authors do not provide the citation of the method that they are likely reproducing (cited as “Redacted” in section 2.4). Can the authors please comment on this?

**Deanonymize Review:**

no

**Paper Type:**

validation/application paper

**Questions To Address In The Rebuttal:**

Can the authors please provide responses to the points (1-6) raised in the weakness section above. The authors can focus on addressing the results section, the corresponding claims and discussions part. Also, it would good if they can clarify on they use calibration of paired bootstrapping as mentioned in point (5) above.

---

### Official Review · Reviewer_oeB1 · 2023-02-07

**Confidence:** 3
**Preliminary Rating:** 4
**Recommendation:** Poster

**Summary:**

The authors provide a compelling study to show that the effect of performance variation between training runs due to stochastic processes such as random data sampling or random weights initialization is non-trivial. There is emphasis that this particularly is more relevant for biomedical image analysis competitions. The authors test their hypothesis by retraining established deep-learning models for diagnostic and segmentation tasks. The test shows substantial variance in performance over different training runs.

**Strengths:**

The question about reproducibility of training methods is conveyed well.
The paper is relatively well written and moderately easy to follow.
The experiment design is clear.
Inclusion of the "Challenge Winner Simulation" is an interesting addition to the paper. It establishes the scope of the paper.


**Weaknesses:**

The motivation of the paper is not strongly established early on. Discussions about clinical significance and the scope of biomedical image analysis leader boards is lacking a bit to drive the motivation.

The range/method/amplitude of variations of the "stochastic" factors has not been established clearly.




**Deanonymize Review:**

no

**Paper Type:**

both

**Questions To Address In The Rebuttal:**

Overall, the paper is moderately well-written. Clarifying the clinical significance and scope of the variance for evaluation of leader board models early can make it flow better(and establish the motivation stronger).

---

### Meta-Review · Area_Chair_nkm8 · 2023-02-22

**Recommendation:** Accept (Poster)
**Confidence:** 4

**Metareview:**

The paper is a study of (ir)reproducibility of deep learning methods for medical imaging, caused only by stochastic processes during training. The results on a range of segmentation and classification tasks show substantial variation, which is currently often not accounted for in practice (e.g. medical imaging competitions).

The reviewers agree on the clarity of the paper as well the quality of the analysis. Some clarifications were raised by the reviewers, these were answered by authors in the rebuttal, and in turn acknowledged by the reviewers. I'm happy to recommend acceptance of the paper.